# Nutrient Effect on the Taste of Mineral Waters: Evidence from Europe

**DOI:** 10.3390/foods9121875

**Published:** 2020-12-16

**Authors:** Vladimir Honig, Petr Procházka, Michal Obergruber, Hynek Roubík

**Affiliations:** 1Department of Chemistry, Faculty of Agrobiology, Food and Natural Resources, Kamýcká 129, 165 00 Prague 6, Czech Republic; honig@af.czu.cz (V.H.); obergruber@af.czu.cz (M.O.); 2Department of Economics, Faculty of Economics and Management, Czech University of Life Sciences Prague, Kamýcká 129, 165 00 Prague 6, Czech Republic; pprochazka@pef.czu.cz; 3Department of Sustainable Technologies, Faculty of Tropical AgriSciences, Czech University of Life Sciences Prague, Kamýcká 129, 165 00 Prague 6, Czech Republic

**Keywords:** mineral water, sensory analysis, consumer preference, atomic absorption spectroscopy, mineral nutrients

## Abstract

In this study, 15 selected bottled mineral waters from chosen European countries were tested for their mineral nutrient contents. In particular, six important nutrients (Ca^2+^, Mg^2+^, Na^+^, K^+^, ${\mathrm{HCO}}_{3}^{-}$, Cl^−^) were measured using atomic absorption spectroscopy. The content of mineral nutrients in all sampled mineral waters were compared to their expected content based on the label. Consequently, their taste was evaluated by 60 trained panelists who participated in the sensory analysis. The results from both the atomic absorption spectroscopy and sensory analysis were analyzed using the regression framework. On the basis of the results from the regression analysis, we determined to what extent the individual mineral nutrients determined the taste of the mineral water. According to the regression results, four out of six analyzed nutrients had a measurable impact on taste. These findings can help producers to provide ideal, health-improving nutrients for mineral water buyers.

## 1. Introduction

The body of an average adult male is 60% (*w*/*w*) water and the body of an average adult woman is 55% (*w*/*w*). There may be significant differences between individuals on the basis of many factors such as age, health, weight, and gender. Body water is divided into intracellular and extracellular fluids. The intracellular fluid, which makes up about two-thirds of the body water, is the fluid contained in the cells. The extracellular fluid that makes up one-third of the body’s fluids is the fluid contained in the areas outside the cells. Extracellular fluid itself is divided into plasma (20% extracellular fluid), interstitial fluid (80% extracellular fluid), and transcellular fluid, which is normally ignored in water calculations, including gastrointestinal, cerebrospinal, peritoneal, and ocular fluid [1].

The amount of water should one drink is broadly discussed outside of scientific circles, but there is no one-size-fits-all answer. The daily four-to-six cup rule is for generally healthy people. This can also vary, especially if there is a water loss through sweat because of exercising or higher temperature [2]. Drinking water is usually consumed as bottled water or as tap water. Bottled waters are generally very popular and also very diverse in terms of overall mineral content and composition. There are many mineral spring drinking waters on the market, which are much diversified in terms of mineral composition. 

According to Act 275/2004 Coll. §2 of Czech law, bottled infant water is a product of high-quality water from a protected underground source, which is suitable for the preparation of infant food and for permanent direct consumption by all members of the population. The total mineral content may not exceed 500 mg/L. As this treatment prohibits any alteration of its composition, infant water is the only bottled water for which the original natural composition is guaranteed [3].

According to the same law, bottled spring water is a product of quality water from a protected underground source that is suitable for permanent direct consumption by children and adults. The total mineral content may not exceed 1000 mg/L (as in the case of drinking water) and the water may only be treated by the physical means mentioned above. The term “spring water” replaced the former “table water”. No substances other than carbon dioxide may be added to packaged infant or spring water [3].

Bottled drinking water is a product that meets the drinking water requirements. This water can be obtained from any water source and treated the same way as tap water, with the quality requirements being similar. In contrast to the above-mentioned types of bottled water, bottled *drinking* water can be artificially supplemented with minerals (calcium, magnesium, sodium, and potassium). If this happens, the list of supplemented substances in the water and the word sign “artificially supplemented with mineral nutrients—mineralized drinking water” should also be included on the label. Bottled drinking water can also be carbonated. Bottled drinking water is marketed under different names (besides trademarks it is, e.g., sparkling water or table water), but it must always be stated on the label that it is drinking water [4].

### 1.1. Mineral Nutrients

The mineral nutrient contents are important characteristics of mineral water. Mineral nutrients are inorganic substances that must be ingested and absorbed in adequate amounts to satisfy a wide variety of essential metabolic and/or structural functions in the body [5]. Mineral water contains a combination of the main cations (Ca^2+^, Mg^2+^, Na^+^, K^+^), anions (${\mathrm{HCO}}_{3}^{-}$, Cl^−^), and specific compounds (which can determine the medicinal value of water) in varying amounts 

All mineral nutrient contents can be read from individual labels provided on the packaging. Labeling fulfils Directive 2009/54/EC of The European Parliament and of The Council on the exploitation and marketing of natural mineral waters from 18 June 2009 [6]. Mandatory criteria that have to be written on a label are presented in Table 1. Overview of mineral nutrients and their intake as well as dislike thresholds are presented in Table 2.

#### 1.1.1. Calcium

The equilibrium state of calcium is given by the relationship among its intake and its absorption and excretion. Even small changes in absorption and excretion can neutralize its high intake or compensate for the lack. There are large variations in calcium intake across countries, depending mainly on milk and dairy consumption. Developing countries have the lowest consumption, especially in Asia, and the highest consumption is found in Europe and North America [7]. 

#### 1.1.2. Magnesium

Most magnesium is found in bones and teeth. To a lesser extent, it is found in the blood and tissues. It is irreplaceable for most biochemical reactions in the human body and its deficiency worsens the course of virtually every disease. We take magnesium by eating bananas, nuts, or almonds. Magnesium relieves irritability and nervousness, releases energy from glucose (blood sugar), and affects proper bone structure. It keeps the circulatory system in good condition and prevents heart attacks. Deficiency can be caused by increased consumption of alcoholic beverages, caffeinated beverages, consumption of semi-finished products, stressful situations, and consumption of certain drugs such as various antibiotics or contraceptives.

#### 1.1.3. Sodium

Virtually all sodium present in food and water is rapidly absorbed by the gastrointestinal tract. Its level in extracellular fluids is maintained by the kidneys and determines their volume. Sodium balance is controlled through a complex mechanism involving both the nervous and hormonal systems. Sodium is mainly excreted in the urine in amounts that correlate with dietary intake [8].

It is also important in maintaining acid–base balance and thus pH in the human body; it also contributes to the control of blood pressure. Sodium and potassium are minerals necessary for the function of muscles and nerves, and potassium plays a vital role in the heart. Sodium-rich mineral waters can be an adjunct to conditions where excessive sweating occurs but are not suitable for long-term consumption for people suffering from hypertension and chronic heart disease. Sodium is hardly present in drinking tap water [9,10].

#### 1.1.4. Potassium

The adult human body contains a total of approximately 110–137 g of potassium, with 98% stored inside the cells, and only 2% in the extracellular fluid [11]. Potassium is the cation most commonly found in intracellular fluid and ensures proper potassium distribution across the cell membrane, which is essential for normal cell function. Long-term maintenance of potassium homeostasis is ensured by changes in its renal excretion as a function of changes in its intake [7]. It is one of the most abundant minerals in the human body. In unprocessed foods, potassium is most often present in the form of bicarbonate generators such as citrate. The potassium added during the process is mostly potassium chloride. The body absorbs about 85% of the intake of potassium [10].

#### 1.1.5. Chlorine

Chlorine exists primarily in the form of sodium chloride in nature [11]. The electrolyte balance is maintained in the body by adjusting the excretion of the kidneys and the gastrointestinal tract according to their total intake. In healthy individuals, chloride is almost completely absorbed in the proximal part of the small intestine. Normal fluid loss is about 1.5 to 2 L per day, along with about 4 g of chloride per day. Most (90–95%) are excreted in the urine, while less is excreted in the feces (4–8%) and sweat (2%) [12]. Healthy individuals can tolerate large amounts of chloride, provided there is sufficient concomitant intake of freshwater. Hypochloremia (a decrease of Cl^−^, especially in extracellular fluid) leads to a decrease in glomerular filtration in the kidneys.

#### 1.1.6. Neutralization Capacity and ${\mathrm{HCO}}_{3}^{-}$

For natural, drinking, and service water, it is assumed that the most important buffer system is carbon dioxide (free)–bicarbonate–carbonate. It can be said that at approximately pH 4.5, all total carbon dioxide will be present in the form of free carbon dioxide, and at a pH of about 8.3, all total carbon dioxide will be in the form of bicarbonates. Normal water is usually in the range of 4.5–8.3; therefore, acid neutralizing capacity (ANC) is usually determined to pH 4.5 using a standard solution of hydrochloric acid and base neutralizing capacity (BNC) to pH 8.5 is determined using a sodium hydroxide solution [13].

## 2. Literature Review

The World Health Organization (WHO) has issued recommendations to define criteria of comfort and pleasure (water pleasant to drink, clear, and with a balanced mineral content). These recommendations are the basis used by the European Union to prepare directives to define drinking water as lacking any particular taste [30]. Nevertheless, different waters have different tastes. This is caused by dissolved minerals in the water. 

Cations such as calcium, sodium, and potassium impact drinking water taste. A neutral taste is encountered where CaCl_2_ < 120 mg/L and Ca(HCO_3_)_2_ > 610 mg/L, although when CaCl_2_ is at levels > 350 mg/L, water is disliked. The optimum sodium concentration is 125 mg/L for distilled water and is typically found as NaHCO_3_ and Na_2_SO_4_. Water is disliked when NaHCO_3_ exceeds 630 mg/L and > 75 mg/L Na_2_CO_3_. Potassium is typically present at low levels as KHCO_3_, K_2_SO_4_, and KCl, and is important at the cellular level of the taste buds. A low potassium concentration has positive effects on water acceptance. KCl acts similar to NaCl in taste effects. Magnesium is typically present in water as MgCO_3_, Mg(HCO_3_)_2_, MgSO_4_, and MgCl_2_ and can impart an astringent taste, being able to be tasted at 100–500 mg/L [24]. 

Water containing magnesium salts at 1000 mg/L has been considered acceptable [31]. Consumers dislike water containing MgCl_2_ > 47 mg/L and Mg(HCO_3_)_2_ > 58 mg/L. Anions such as bicarbonate, chloride, and sulfate also impact the taste. At neutral pH, the bicarbonate is more common than carbonate and helps keep cations in solution. In contrast, carbonate increases at higher pH and at lower dissolved CO_2_ levels. Aeration also adds O_2_ and removes CO_2_, promoting carbonate precipitation. The taste threshold concentration for chloride is 200–300 mg/L [24,27]. Increased chloride levels in the water in the presence of sodium, calcium, potassium, and magnesium can cause water to become objectionable. Preference testing has revealed that water containing NaCl < 290 mg/L is acceptable and NaCl > 465 mg/L is disliked. Testing also indicates that CaCl_2_ < 120 mg/L is neutral, while CaCl_2_ > 350 mg/L is disliked [18].

Sensory analysis of water can be conducted for example by Quantitative Descriptive Analysis [32] or Quantitative Flavor Profiling [33]. However, a detailed methodology was discussed earlier in Krasner et al. [34] and Suffet et al. [35], resulting in the standard method AWWA [36]. There is also a number of standards and articles focused only on odor [35,37,38,39]. In one of the recent articles, Rey-Salgueiro et al. [40] evaluated bottled natural mineral water (17 still and 10 carbonated trademarks) to propose training procedure for new panelists. The tasting questionnaire included 13 attributes for still water plus overall impression, and they were sorted by color hues, transparency and brightness, odor/aroma, and taste/flavor/texture, and two more for carbonated waters (bubbles and effervescence). 

Harmon et al. [41] investigated people’s preferences for different water sources and factors that predict such preferences using a blind taste test. Water preferences of 143 participants for one name-brand bottled water, one groundwater-sourced tap water, and one indirect potable reuse (IDR) water were assessed. For predictors of water preference, we measured each participant’s phenylthiocarbamide taste sensitivity.

Koseki et al. [42] evaluated taste of alkali ion water (calcium sulfate or calcium lactate added to tap water with a calcium concentration of 17.5 mg/L, creating a calcium concentration of 40 mg/L) and bottled mineral waters. There were two studies, one with 166 panelists and one with 150 panelists. For evaluation of taste, the studies used a five-point Likert scale. The goal was to assess improvement or deterioration of taste after adding alkali. It was found that addition is preferable in any mineral water.

Platikanov et al. [28,43] conducted chemometric analysis experiments with sensory analysis of water samples. The first experiment was divided into two studies. These were conducted with 17 and 13 trained panelists and 23 and 28 samples of water, respectively. The second experiment was conducted with 69 untrained volunteers and 25 samples of water. Both cases used bottled and tap water. In both cases, the researchers was concluded that the most important factor that influenced panelists’ preferences was the overall level of mineralization. Water samples with high levels of mineralization were rated with low scores.

All aforementioned mineral nutrients also influenced the taste of mineral water. Consumers have different preferences as to which mineral water they will buy. One of these criteria is necessarily the taste of mineral water. Taste of mineral water in terms of nutrients is, therefore, one of the most important characteristics that determine the success of mineral waters in the market. Similarly, it is important to determine which mineral nutrients are perceived by consumers to be tasty and whether or not poor taste may make consumers omit consumption of important health-supporting nutrients from mineral water. 

In this paper, we examined six important nutrients and their influence on the taste of water. Firstly, the nutrient content in water was measured, focusing on mineral waters of generally >500 mg/L mineral residue or TDS. Then, a model was built to define and explain their influence on taste. Finally, on the basis of the findings, conclusions were drawn.

## 3. Materials and Methods

### 3.1. Determination of Composition

Three samples of each mineral water available on the market in Central Europe were analyzed. 

${\mathrm{HCO}}_{3}^{-}$ determination: 100 mL of the sample was collected in a titration flask and 3 drops of methyl orange were added. Then, the sample was titrated with a standard hydrochloric acid solution until the first hint of onion coloring occurred, the value of acid consumed was recorded. Three measurements were made for each sample. The bicarbonate concentration was calculated using Equation (1) below.
(1)$$C_{{\mathrm{HCO}}_{3}^{-}}=\frac{C_{\mathrm{HCl}}\cdot V_{\mathrm{HCl}}}{V}\cdot M_{{\mathrm{HCO}}_{3}^{-}}$$
where 

$C_{{\mathrm{HCO}}_{3}^{-}}$ is the molar concentration of ${\mathrm{HCO}}_{3}^{-}$ (mol/L^−1^);

$C_{\mathrm{HCl}}$ is molar concentration of HCl (mol/L^−1^);

$V_{\mathrm{HCl}}$ is the volume of HCl (L);

V is the volume of water (L);

$M_{{\mathrm{HCO}}_{3}^{-}}$ is the molar weight of ${\mathrm{HCO}}_{3}^{-}$ (g·mol^−1^).

Determination of Cl^−^: A total of 100 mL of the sample was collected in a titration flask and 1 mL of potassium dichromate was added. The sample was then titrated with a standard solution of AgNO_3_ (0.07372 mol/L^−1^) until the first constant red color occurred. The value of the consumed solution was recorded and calculated according to Equation (2) below. The volumetric solution itself was standardized with sodium chloride base.
(2)$$C_{{\mathrm{Cl}}^{-}}=\frac{V_{{\mathrm{AgNO}}_{3}}\cdot C_{{\mathrm{AgNO}}_{3}}}{V}\cdot M_{\mathrm{Cl}}$$
where

$C_{{\mathrm{Cl}}^{-}}$ is molar concentration of Cl (mol·L^−1^);

$C_{{\mathrm{AgNO}}_{3}}$ is the molar concentration of AgNO_3_ (mol·L^−1^);

$V_{{\mathrm{AgNO}}_{3}}$ is the volume of AgNO_3_ (L);

V is the volume of water (L);

$M_{{\mathrm{Cl}}^{-}}$ is the molar weight of Cl (g·mol^−1^).

Other elements were measured by AAS (atomic absorption spectroscopy). The VARIAN SPECTR AA 110 Atomic Absorption Spectrometer (GFAAS, Varian AA280Z, Varian, Australia equipped with a GTA-110Z graphite furnace atomizer) was used for measurement (Figure 1).

For measurement of calcium, a calcium carbonate standard (1 g/L) was used; magnesium standard—magnesium carbonate (1 g/L); sodium standard—sodium carbonate (1 g/L); potassium standard—potassium nitrate (1 g/L); 1.5% nitric acid; 5% lanthanum chloride; and demineralized water.

For measurements with AAS, an acetylene/air gas mixture was used, and it was measured with a hollow discharge cathode for the particular elements. After switching off the gas, heating the lamp, and optimizing the source, we first measured the standard in order to determine the calibration curve. Samples were then measured, and a blank of demineralized water was measured between each of the 3 samples, with the instrument reporting results in mg/L concentrations, from which the average of the blank was subtracted and converted to the original concentration before dilution.

### 3.2. Sensory Analysis

#### 3.2.1. Grouping

Refreshing, tasty, and good water is formed by a certain (optimal) concentration of inorganic water components. It is significantly affected mainly by concentrations of calcium, magnesium, iron, manganese, bicarbonates, alumina, etc. Since there are clear differences in mineral waters, they were divided into several groups. The first group is based on mineralization. Waters were split into 4 groups: VPM = very poorly mineralized, with the dissolved substances up to 50 mg/L;PM = poorly mineralized, with the content of solutes 50 to 500 mg/L;MM = moderately mineralized, with the content of solutes 500 to 1500 mg/L;SM = strongly mineralized, with the content of solutes 1500 to 5000 mg/L;VSM = very strongly mineralized, with the content of solutes greater than 5000 mg/L.

The second group is based on pH:base;neutral;acidic.

The third group is based on country of origin.

Finally, we anonymized all brands by all of these groups and numbered them.

#### 3.2.2. Panelists, Water Samples, and Preparation

The taste and smell of water are among the most important characteristics of drinking water quality from the consumer’s point of view. Subjective sensory evaluation, therefore, has its irreplaceable place in the monitoring of drinking water quality. To evaluate the taste of individual mineral waters, 60 trained panelists participated in the sensory analysis of taste. 

The sensory water analysis training is carried out by the National Institute of Public Health (NIPH).

Determination of the smell and taste of mineral waters was performed by experts who completed a course in sensory analysis and according to the standard EN 1622 standard. This standard defines sampling, test environment, standardized procedure, and expression of results.

Each panelist was always randomly assigned 1 water from each group to assess the full range of samples. Members worked individually and no discussion took place after the session. Samples were administered at room temperature in 200 mL beakers filled to approximately one-third of their volume. All samples were served at room temperature (20 °C). 

Samples were labeled with a 4-digit code and the samples were poured in a different room to avoid influencing the rating due to the personal preferences of the evaluators.

Since each evaluator evaluated only 5–7 samples (18 evaluators per water), the waters were divided into 5 groups according to the amount of minerals with 3 samples, according to the data of total dissolved content.

#### 3.2.3. Descriptive Sensory Analysis

The evaluators assessed the overall taste on a 10 cm graphic unstructured scale with a border of both sides, in which one extreme was marked “zero taste intensity (1)” and the other as “maximum taste intensity (10)”. The graphical representation on the scale was converted to values with a ruler to the nearest half-centimeter.

### 3.3. Regression Analysis

Regression analysis was performed [44,45,46]. Using regression analysis, we were able to obtain information as to whether, for example, the higher concentration of chloride in mineral water improved taste or, on the other hand, the higher concentration of bicarbonate deteriorated taste. 

The general model for *n* variables is in the form of Equation (3):(3)$$y=\beta_{0}+\beta_{1}x_{1}+\beta_{2}x_{2}+\ldots +\beta_{n}x_{n}+e$$
where

*y* is a vector of dependent variables;

*β* is a vector of regression coefficients;

*x* are vectors for independent variables;

*e* is a matrix associated with errors of the estimation. 

On the basis of the results of sensory analysis results, we performed linear regression analysis between taste as the dependent variable and individual nutrients as independent variables. All statistical analyses were conducted in statistical software R version 3.6.1 [47].

## 4. Results

### 4.1. Atomic Absorption Spectrometry Results

Results from the mineral nutrients measurements are presented in Table 2. Authors grouped mineral waters according to the methodology presented above. Results from Table 3 are compared to measurements on the label.

There were no statistically significant differences (in terms of paired *t*-test) between the data measured on AAS and the data on the labels (Table 3). Minor deviations can be expected, since the evaluation of water on the labels is usually not up to date, as their analyzes are carried out at larger intervals and the quantitative characteristics of the water may vary depending on many factors such as temperature, rainfall, and potential momentaneous industrial and agricultural pollution [48].

### 4.2. Results of Sensory Analysis and Hedonic Pricing

Sensory analysis was performed on 15 samples where every sample was tested 18 times. Results of the sensory analysis are presented in the last column of Table 2. Finally, on the basis of the results from nutrient content measurement and taste perception, we performed a regression analysis in order to determine a possible relationship between composition and taste. After several iterations where some of the mineral nutrients had to be removed as they were superfluous, we generated the final regression results, which are presented in Table 4. Two removed nutrients are chlorine and potassium. A significant regression was found (*F*(6, 8) = 8.56, *p* < 0.005, *N* = 18), with *R*^2^ of 0.8652. Assumptions of linear regression were verified by R library gvlma [49] and by Breusch–Pagan test of heteroscedasticity [50]. All assumptions of linear regression were accepted: global stats (*p* = 0.5564), heteroscedasticity (*p* = 0.2981), skewness (*p* = 0.1689), kurtosis (*p* = 0.9544), and link function (*p* = 0.8624). 

Results indicate that selected nutrients have a significant impact on taste, albeit their influence is relatively small. For example, when the concentration of calcium goes up by 1 mg per liter, the perceived taste goes up 0.01. A similar interpretation can be made in terms of other mineral nutrients. While calcium and magnesium have a positive impact on taste, sodium and bicarbonate have a negative impact on taste. Magnesium has the relatively strongest influence on taste, while bicarbonate has an influence that is approximately 17 times smaller. It has to be acknowledged, however, that the estimates of coefficients are probably valid within a certain range. Above a certain threshold, the positive effect on taste may cause a negative perception. 

Moreover, the difference between the predicted value and real sensory taste value is depicted in Figure 2. Figure 2 also graphically benchmarks taste of individual mineral waters. 

## 5. Discussion

Many authors who examine taste concentrate primarily on the amount of total dissolved solids (TDS) [51]. TDS can be described as the total amount of dissolved cations Al^3+^, Fe^2+^, Mn^2+^, Ca^2+^, Mg^2+^, K^+^, and Na^+^, and anions such as ${\mathrm{CO}}_{3}^{2-}$, ${\mathrm{HCO}}_{3}^{-}{,\;\mathrm{SO}}_{4}^{2-}$, and Cl^−^. For example, Bruvold and Daniels (1990) [51] claimed that the higher is the amount of TDS, the poorer the taste. Daniels et al. (1988) claimed that a high amount of TDS may lead consumers to refuse to drink water at all [52].

According to Kozisek (2004) [4], cations such as calcium, sodium, and potassium impact drinking water taste. While calcium is perceived usually neutrally, sodium can be perceived to influence the taste of water negatively. Potassium is perceived mainly neutrally, albeit the low potassium level increases acceptance of water.

Magnesium is usually perceived mainly positively [31]. 

For chloride, enhanced chloride levels in mineral water can cause water to become less acceptable.

This is particularly true when sodium, calcium, potassium, and magnesium are present in water [27]. In some studies, chloride was found to be neutral [53]. In other studies, chloride content in water was considered to be negative while its effect can be mitigated by decreasing water temperature [54].

Whelton et al. (2007) [18] provided a summary of mineral effects on the taste of drinking water of Cl^−^, ${\mathrm{HCO}}_{3}^{-}$, Na^+^, K^+^, and Mg^2+^. The review indicates that K^+^ is perceived mainly positively, Ca^2+^ and Na^+^ are perceived both neutrally or positively, and Cl^−^ and Mg^2+^ are perceived neutrally or negatively. Acceptance is usually dependent on concentration of other minerals.

Chidya et al. (2019) [55] published an article wherein they found differences between the labels and the actual water content in some cases, although we did not find this to be the case in this article. Issues were found in terms of pH and concentration of F^−^ anions. 

Zuliani et al. (2020) [56] presented a multi-elemental analysis of 13 bottled waters and found similar results to those in this article in that mostly ${\mathrm{HCO}}_{3}^{-}$, Ca^2+^, and Mg^2+^ dominated in bottled mineral and spring water. 

This paper is novel in terms of taste perception in the selection of the composition of analyzed waters. The drinking waters investigated in this paper mostly contained higher levels of Ca^2+^ and ${\mathrm{HCO}}_{3}^{-}$, while levels of Cl^−^ were lower (compared with [28,41,43,48] by Welch *t*-test). Drinking water considered in these papers was also studied in terms of water concentration of total dissolved solids less than 500 mg/L. In this paper, we also analyzed drinking waters with the sum of minerals above 500 mg/L (in cases of moderately mineralized, strongly mineralized, and healing waters). For these waters, the data also showed that if the concentration of HCO_3_^−^ to concentration of Cl^−^ was greater than 50, Cl^−^ did not significantly (*p*-value < 0.01) affect the taste, and only HCO_3_^−^ was important.

Generally, as tap water flavor is among the major concerns for water supporters, only a minor percentage is used for drinking [48], increasing the importance of bottled water composition and flavor [57]. It needs to be kept in mind that consumers assess their tap water primarily by its initial assessment via taste, odor, as well as appearance [58,59]. However, similar preferences and assessments can be also seen in quality of drinking water produced by for example reverse osmosis [60] or other filtration methods, as well as, for example, when optimizing drinking water taste by appropriate adjusting of mineralization (measured by TDS), such as that done by [61]. In any case, it is important not to forget also about the role of consumer preferences, which are also valid in cases of the mineralization of water, wherein the preference ratings vary [48]. However, it is necessary to uncover taste determinants as it is the initial basis for water industry providers in understanding perceptions and preferences among types of drinking water [48].

Therefore, determination as to what extent the individual mineral nutrients determine the taste of the mineral water is essential, as such findings can help producers to provide ideal, health-improving nutrients for mineral water buyers.

## 6. Conclusions

The measurement of nutrients in bottled mineral water in selected European countries shows that the measured values of nutrients correspond for the most part with the label values. This paper brings about new findings with regards to the influence of mineral nutrients on the taste of mineral water. Results revealed which minerals are considered to be tasty, even though the producers of mineral waters may influence their content in terms of better satisfying consumer preferences. At the same time, given that the health effects of certain minerals are well known, the optimal taste can contribute to the improvement of consumers’ health. Specific nutrients were allocated that affect water in both positive and negative ways. Our study also confirms the findings of other authors, such as in the case of magnesium being perceived positively.

On the other hand, some results were quite surprising, such as chloride being found to have an insignificant impact on taste. The present study provides directions to producers of mineral waters as to how to change the content of certain minerals and in this way better satisfy the needs of their customers. In cases wherein better taste is also associated with improvement of human health due to certain mineral nutrient contents, benefits from drinking mineral water may spill over into the wider society due to higher healthcare cost savings. 

## Figures and Tables

**Figure 1 foods-09-01875-f001:**
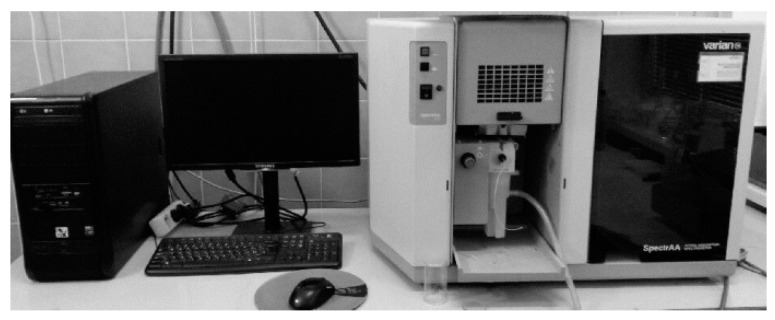
Atomic absorption spectroscopy Varian AA280Z.

**Figure 2 foods-09-01875-f002:**
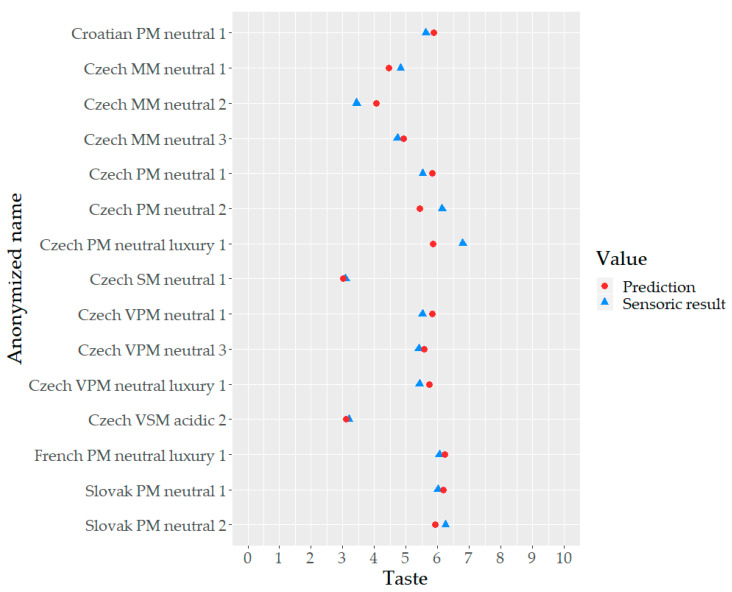
How well the model predicted the actual value of taste. For all but three mineral waters, predicted value based on the estimated model corresponded well with the actual value of taste. Exceptions were Czech MM neutral 2, Czech PM neutral 2, and Czech PM neutral luxury 1. However, even for those three types, the difference was around 15%.

**Table 1 foods-09-01875-t001:** Indications and criteria laid down in article 9(2) of 2009/54/EC [6].

Indications	Criteria
Very low mineral content	Mineral salt content, calculated as a fixed residue, not greater than 50 mg/L
Low mineral content	Mineral salt content, calculated as a fixed residue, not greater than 500 mg/L
Rich in mineral salts	Mineral salt content, calculated as a fixed residue, greater than 1500 mg/L
Contains bicarbonate	Bicarbonate content greater than 600 mg/L
Contains sulphate	Sulphate content greater than 200 mg/L
Contains chloride	Chloride content greater than 200 mg/L
Contains calcium	Calcium content greater than 150 mg/L
Contains magnesium	Magnesium content greater than 50 mg/L
Contains fluoride	Fluoride content greater than 1 mg/L
Contains iron	Bivalent iron content greater than 1 mg/L
Acidic	Free carbon dioxide content greater than 250 mg/L
Contains sodium	Sodium content greater than 200 mg/L
Suitable for the preparation of infant food	—
Suitable for a low-sodium diet	Sodium content less than 20 mg/L
Maybe laxative	—

**Table 2 foods-09-01875-t002:** Overview of mineral nutrients [7,8,9,10,11,12,13,14,15,16,17,18,19,20,21,22,23,24,25,26,27,28,29].

Mineral	Volume	Form in Body	Taste	Intake	Dislike	Threshold
Calcium	1000 g	Ca_10_(PO_4_)_6_(OH)_2_, Ca(HCO_3_)_2_, CaSO_4_, CaCl_2_, Ca_3_SiO_5_, Ca_2_SiO_4_, CaCO_3_	Bitter, sour [20]	Up to 100 mg	Ca(HCO_3_)_2_ > 610 mg/L CaCl2 > 350 mg/L.	100–300 mg/L
Magnesium	25 g	MgCO_3_, Mg(HCO_3_)_2_, MgSO_4_, MgCl_2_, CaMg(CO_3_)_2_	Bitter or salty-bitter	10 mg	MgCl_2_ > 47 mg/L Mg(HCO_3_)_2_ > 58 mg/L.	45–60 mg/L
Sodium	92 g	NaHCO_3_, Na_2_SO_4_, Na_2_CO_3_, NaCl	salty taste	5–20 g	NaHCO_3_ > 630 mg/L Na_2_CO_3_ > 75 mg/L	2 mmol/L
Potassium	110–137 g	KHCO_3_, K_2_SO_4_, KCl, K_2_HPO_4_, K_2_CO_3_	Salty-bitter, salty-alkaline	90 mmol (3510 mg)	NaCl > 465 mg/L	43.3 mmol/L
Chlorine	81.7 g	NaCl, CaCl_2_, KCl, MgCl_2_	Bitter tasting	9 mg/kg	NaCl > 465 mg/L CaCl2 > 350 mg/L	200–300 mg/L
Carbonate, bicarbonate	–	NaHCO_3_, Mg(HCO_3_)_2_, Ca(HCO_3_)_2_, KHCO_3_	Typical mineral tart taste	-	-	180–285 mg/L

**Table 3 foods-09-01875-t003:** Measured parameters of mineral waters in Central Europe.

Anonymized Name	Composition mg·L^−1^						Sum ^1^	Taste
	Ca^2+^	Mg^2+^	Na^+^	K^+^	${\mathrm{HCO}}_{3}^{-}$	Cl^−^		
	**Very Poorly and Poorly Mineralized**							
Czech VPM neutral 1	20.21	2.81	4.86	1.1	70.15	8.49	107.6	5.52
Czech VPM neutral 3	6	8.6	11.3	10.7	111	1.01	148.6	5.4
Czech VPM neutral luxury 1	26.8	3.24	5.47	0.424	110	1.5	147.4	6.79
Czech PM neutral 1	82.35	3.45	1.05	1.07	231.8	5.23	325.0	5.53
Czech PM neutral 2	43.6	14.12	25.7	4.92	213.5	3.02	304.9	6.14
Czech PM neutral luxury 1	76.7	2.75	2.65	1.41	202	1.54	287.1	6.79
French PM neutral luxury 1	94	20	7.7	5	248	4	378.7	6.06
Slovak PM neutral 1	79	38.5	16.4	2.4	317	17	470.3	6.01
Slovak PM neutral 2	87	19.2	2.6	1.1	324	4.8	438.7	6.25
Croatian PM neutral 1	64.2	32.1	1.7	0.6	372.3	2.8	473.7	5.63
	Moderately Mineralized							
Czech MM neutral 1	73.51	20.16	47.41	18.42	469.7	10.45	639.7	4.83
Czech MM neutral 2	84.5	25	69.9	15.01	528	12	734.4	3.44
Czech MM neutral 3	255.78	21.06	26.65	1.48	863.15	7.19	1175.3	4.73
	Strongly Mineralized							
Czech SM acidic 1	426	128	10.8	17.5	1763	34.7	2380.0	3.09
	Healing (Very Strongly Mineralized)							
Czech VSM acidic 2	279	143	93.5	89	1600	39.3	2243.8	3.19

^1^ Sum of measured minerals.

**Table 4 foods-09-01875-t004:** Hedonic analysis of taste.

Term	Estimate	*p*-Value	Significance ^1^
(Intercept)	5.937	1.2·10^−10^	***
Ca^2+^	0.010	0.015	*
Mg^2+^	0.028	0.054	.
Na^+^	−0.017	1.4·10^−3^	**
${\mathrm{HCO}}_{3}^{-}$	−0.004	3.6·10^−3^	**

^1^ Significance sign qualitatively evaluates *p*-value. If the *p*-value is between 0 and 0.001, the significance is ***. Similarly, between 0.001 and 0.01 is **, between 0.01 and 0.05 is *, and 0.05 and 0.1 is “.”.

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
