# Peer review of "Nutrient Effect on the Taste of Mineral Waters: Evidence from Europe"

_foods, 2020, doi:10.3390/foods9121875_

Round 1

Reviewer 1 Report

This manuscript describes the correlation between targeted minerals in mineral water and the overall liking of these water samples. The authors made an effort to present the data collected in a logical manner with some statistical analysis, but I feel that the content is lacking depth and requires more discussion.  Here is a list of things I would recommend doing:

  • The introduction needs to provide more relevant information and be succinct. The Introduction was about half the length of the manuscript  and a significant portion of it was explaining nutrient metabolism of various minerals, which is irrelevant to the title and objective of this study.
  • The sensory analysis needs to be more descriptive to develop depth to the analysis and discussion. The only sensory data reported was an "overall taste" score. Furthermore, even though the panelists were described as "trained", there were no explanation on how these panelists were trained. The scale being used to grade "overall taste" seems very subjective, and it can be influenced by a person's background, which can be varied or controlled (but not mentioned in this manuscript). If a true descriptive sensory analysis was conducted, the results would have provided a richer discussion for this manuscript, which is currently lacking content
  • The "Discussion" portion needs to provide more discussion of the results. The author mainly describes what other literature have found. More references to the results obtained need to be made. For example, how does your result compare to the results from past literature? Does the data make sense? Anything surprising? If so, explain.
  • Also, the anonymized name in Table 2 is confusing. What does neutral luxury mean?

Author Response

Reviewer 1

This manuscript describes the correlation between targeted minerals in mineral water and the overall liking of these water samples. The authors made an effort to present the data collected in a logical manner with some statistical analysis, but I feel that the content is lacking depth and requires more discussion.  Here is a list of things I would recommend doing:

Dear Reviewer 1,

Firstly, let me thank you for all the work you did reviewing our paper. We highly appreciate that.

As for this paper we received comments from 3 reviewers, you can see all the changes done in the paper in track mode. Even though that the majority of comments were of a minor character, for incorporating changes appropriately, we took 2 weeks.

You can see our responses below in green.

Thank you again.

Authors

  • The introduction needs to provide more relevant information and be succinct. The Introduction was about half the length of the manuscript  and a significant portion of it was explaining nutrient metabolism of various minerals, which is irrelevant to the title and objective of this study.

Has been shortened.

  • The sensory analysis needs to be more descriptive to develop depth to the analysis and discussion. The only sensory data reported was an "overall taste" score. Furthermore, even though the panelists were described as "trained", there were no explanation on how these panelists were trained. The scale being used to grade "overall taste" seems very subjective, and it can be influenced by a person's background, which can be varied or controlled (but not mentioned in this manuscript). If a true descriptive sensory analysis was conducted, the results would have provided a richer discussion for this manuscript, which is currently lacking content.

 Information about sensory analysis has been added

  • The "Discussion" portion needs to provide more discussion of the results. The author mainly describes what other literature have found. More references to the results obtained need to be made. For example, how does your result compare to the results from past literature? Does the data make sense? Anything surprising? If so, explain.

Discussion has been enhanced.

  • Also, the anonymized name in Table 2 is confusing. What does neutral luxury mean?

This is used to distinguish among different anonymized types of mineral water and is related to pricing of mineral waters.

Reviewer 2 Report

Although the paper bring originality, it must be improved:

1) introduction and literature review: why is that? It it very extensive. Some of the information can be included in the "discussion" section;

2) Methodology should be more clearly presented:

2.1) "Samples of mineral water available on the market in Central Europe were analyzed" - how many samples for each type of water were tested?

2.2) There are useless information - "Regression analysis is a statistical process done to study the relationship between a set of independent variables (explanatory variables) and the dependent variable (response variable)."

3) In the results section it is not clear if table 2 presents results obtained from research or in food label; additionally, there is some results discussion that mus be integrated in "discussion" section;

4) Discussion section seems incomplete.

Author Response

Reviewer 2

Dear Reviewer 2,

Firstly, let me thank you for all the work you did reviewing our paper. We highly appreciate that.

As for this paper we received comments from 3 reviewers, you can see all the changes done in the paper in track mode. Even though that the majority of comments were of a minor character, for incorporating changes appropriately, we took 2 weeks.

You can see our responses below in green.

Thank you again.

Authors

Although the paper bring originality, it must be improved:

1) introduction and literature review: why is that? It it very extensive. Some of the information can be included in the "discussion" section;

Thank you for the comment, we followed it and we made introductions section more concise.

2) Methodology should be more clearly presented:

Yes, we tried to write it in a more coherent way, which should enable readers easier orientation.

2.1) "Samples of mineral water available on the market in Central Europe were analyzed" - how many samples for each type of water were tested?

Added. We analysed three samples per each mineral water.

2.2) There are useless information - "Regression analysis is a statistical process done to study the relationship between a set of independent variables (explanatory variables) and the dependent variable (response variable)."

Changes and rewritten in a more concise way.

3) In the results section it is not clear if table 2 presents results obtained from research or in food label; additionally, there is some results discussion that mus be integrated in "discussion" section;

Table 2 renamed: Table 2. Measured parameters of mineral waters in Central Europe

4) Discussion section seems incomplete.

We tried to improve the discussion section.

Reviewer 3 Report

Honig et al. Nutrient effects on the taste of mineral waters: Evidence from Europe 

The interesting manuscript combines nutrition and water quality in a novel manner. With edits and more recent references, the manuscript will be improved. 

Lines 66-72. Explain mineral content in the context of Total Dissolved Solids (TDS). TDS is the international criteria for drinking water and its standards and is the basis of the EC criteria listed in Table 1. In addition to the mineral cations, mention the typical anions that are in water which include chloride, sulfate, and carbonates. 

Lines 91-94. Add sulfate as an anion with bicarbonate and chloride. The example of “chloride sulfate” is incorrect. Chloride and sulfate are both anions and do not form a salt. 

Line 120. The word “income” does not make sense. Could this be “consumption”? 

Line 227 and associated paragraph. Add a comment about speciation of chlorine/chloride in the text and clarify which species is being discussed in the body. Sometimes it is unclear if chlorine is present in the body, and if it is, clarify is molecular chlorine or organochlorine. 

Table 1. Rearrange Indicators in value order. Very low mineral content should be first as it is the lowest mineral content in the Table.  

Literature Review 

Line 264 and forward. Literature review. The authors include excellent mineral and taste references from the 20th century; some key works from the 21st century are missed and should be cited as they provide insight to the taste of water and interpreting the data. 

Teillet, E., Urbano, C., Cordelle, S., Schlich, P. 2010. Consumer perception and preference of bottled and tap water. Journal of Sensory Studies, 25(3), 463–480. 

Marcussen, H., Holm, P.E., Hansen, H.C.B. 2013. Composition, flavor, chemical foodsafety, and consumer preferences of bottled water. Comprehensive Reviews in Food Science and Food Safety, 12, 333-352 

García, V., Fernández, A., Medina, M., Ferrer, O., Cortina, J., Valero, F. , Devesa, R. 2014 Flavour assessment of blends between desalinated and conventionally treated sources. Desalination and Water Treatment, 53, 3466-3474. 

Dietrich, A.M., Burlingame, G.A. 2015 Critical review and rethinking of USEPA secondary standards for maintaining consumer acceptability of organoleptic quality of drinking water; Environmental Sciences and Technology, 49(2), 708–720. DOI: 10.1021/es504403t. 

Vingerhoeds, M.H., Nijenhuis-de Vries, M.A., Ruepert, N., van der Laan, H., Bredie, W.L., Kremer, S. 2016 Sensory quality of drinking water produced by reverse osmosis membrane filtration followed by remineralisation. Water Research, 94, 42-51. 

Devesa, R., Dietrich, A. 2018. Guidance for optimizing drinking water taste by adjusting mineralization as measured by total dissolved solids (TDS). Desalination, 49, 147-154, 2018.     https://doi.org/10.1016/j.desal.2018.04.017 

Dietrich, A., Devesa, R. Chapter 8: Characterization and removal of minerals that cause taste. In: Taste and Odour in Source and Drinking Water: Causes, Controls, and Consequences, Editors: T-F. Lin. IWA Publishing, UK. ISBN13: 9781780406657; eISBN: 9781780406664, 2019, pp 245-280. 

Line 303. Define alkali ion and provide chemical structure. 

Line 314-323. Re-state emphatically that this study is focused on mineral waters of generally >500 mg/L mineral residue or TDS (see Table 2) and that many waters were very high in calcium carbonate. This is important for interpretation of the data. 

Methods 

Line 328 and forward. Define the hedonic scale: is 1 a like or dislike?  

Results  

Table 2. Remove “very poorly” as header as no waters taste-tested have < 50 mg/L mineral residue. There are not waters in this group. 

Table 2. While the authors did not measure TDS nor sulfate (a major TDS ion in some waters), it would be helpful for the reader to have a sense of TDS. A suggestion is to add a column to table 2 that is the “Sum of measured Minerals” which will provide an indicator of the overall mineral content of the waters. This will not be TSD, but it will provide perspective. If the water producer/bottler provided TDS as a parameter on their product, then provide the value. 

Discussion 

Summarize data in Table 2 to indicate that most of the high mineral content waters in this study had substantial calcium and carbonate and little chloride. This is what distinguishes this study from other drinking water taste investigations and provides novelty. Most municipal drinking waters strive for TDS< 500 mg/L; these waters are typically >500 mg/L TDS. Chloride<

Lines 458-460. An important aspect of consumer behavior toward drinking water is their perception of the water relative to what they typically drink and their liking of mineral tastes. Consumers like similar and familiar products; Teillet et al. address this. The role of consumer preference in the mineralization of water is similar to that of other consumer products: for example, some consumers like crackers with high salt (NaCl) and a salty taste, others like low salt and less salty taste. 

Line 471. Whelton et al. 2007 is not the correct reference as their article is a review and does not have experimental data. 

Conclusions 

The conclusions are very general and can be improved by added specific and number values. 

Author Response

Reviewer 3

Honig et al. Nutrient effects on the taste of mineral waters: Evidence from Europe 

The interesting manuscript combines nutrition and water quality in a novel manner. With edits and more recent references, the manuscript will be improved. 

Dear Reviewer 3,

Firstly, let me thank you for all the work you did reviewing our paper. We highly appreciate that.

As for this paper we received comments from 3 reviewers, you can see all the changes done in the paper in track mode. Even though that the majority of comments were of a minor character, for incorporating changes appropriately, we took 2 weeks.

You can see our responses below in green.

Thank you again.

Authors

Lines 66-72. Explain mineral content in the context of Total Dissolved Solids (TDS). TDS is the international criteria for drinking water and its standards and is the basis of the EC criteria listed in Table 1. In addition to the mineral cations, mention the typical anions that are in water which include chloride, sulfate, and carbonates. 

Thank you for comment. Added.

Lines 91-94. Add sulfate as an anion with bicarbonate and chloride. The example of “chloride sulfate” is incorrect. Chloride and sulfate are both anions and do not form a salt. 

Added: Anions such as bicarbonate, chloride, and sulfate also impact the taste.

Line 120. The word “income” does not make sense. Could this be “consumption”? 

Agreed, thank you for suggestion. Developing countries have the lowest consumption, especially in Asia and the highest in Europe and North America

Line 227 and associated paragraph. Add a comment about speciation of chlorine/chloride in the text and clarify which species is being discussed in the body. Sometimes it is unclear if chlorine is present in the body, and if it is, clarify is molecular chlorine or organochlorine. 

We have rewritten paragraph accordingly.

1.1.5. Chlorine

Chlorine exists primarily in the form of sodium chloride in nature [18]. The electrolyte balance is maintained in the body by adjusting the excretion of the kidneys and the gastrointestinal tract according to their total intake. In healthy individuals, chloride is almost completely absorbed in the proximal part of the small intestine. Normal fluid loss is about 1.5 to 2 liters per day, along with about 4 g of chloride per day. Most (90–95%) are excreted in the urine, less in the feces (4–8%) and in sweat (2%) [23]. Healthy individuals can tolerate large amounts of chloride provided there is sufficient concomitant intake of freshwater. Hypochloremia (a decrease of Cl- especially in extracellular fluid) leads to a decrease in glomerular filtration in the kidneys.

Table 1. Rearrange Indicators in value order. Very low mineral content should be first as it is the lowest mineral content in the Table.  

Done.

Literature Review 

Line 264 and forward. Literature review. The authors include excellent mineral and taste references from the 20th century; some key works from the 21st century are missed and should be cited as they provide insight to the taste of water and interpreting the data. 

Thank you for suggestions, we incorporated majority of them. Except García et al. (2014) as we did not see it fitting and Dietrich and Devesa (2019), as we had no access to the full text.  Otherwise, we incorporated all the recommendations and they served in improvement of the text.

Teillet, E., Urbano, C., Cordelle, S., Schlich, P. 2010. Consumer perception and preference of bottled and tap water. Journal of Sensory Studies, 25(3), 463–480. 

Marcussen, H., Holm, P.E., Hansen, H.C.B. 2013. Composition, flavor, chemical foodsafety, and consumer preferences of bottled water. Comprehensive Reviews in Food Science and Food Safety, 12, 333-352 

García, V., Fernández, A., Medina, M., Ferrer, O., Cortina, J., Valero, F. , Devesa, R. 2014 Flavour assessment of blends between desalinated and conventionally treated sources. Desalination and Water Treatment, 53, 3466-3474. 

Dietrich, A.M., Burlingame, G.A. 2015 Critical review and rethinking of USEPA secondary standards for maintaining consumer acceptability of organoleptic quality of drinking water; Environmental Sciences and Technology, 49(2), 708–720. DOI: 10.1021/es504403t. 

Vingerhoeds, M.H., Nijenhuis-de Vries, M.A., Ruepert, N., van der Laan, H., Bredie, W.L., Kremer, S. 2016 Sensory quality of drinking water produced by reverse osmosis membrane filtration followed by remineralisation. Water Research, 94, 42-51. 

Devesa, R., Dietrich, A. 2018. Guidance for optimizing drinking water taste by adjusting mineralization as measured by total dissolved solids (TDS). Desalination, 49, 147-154, 2018.     https://doi.org/10.1016/j.desal.2018.04.017 

Dietrich, A., Devesa, R. Chapter 8: Characterization and removal of minerals that cause taste. In: Taste and Odour in Source and Drinking Water: Causes, Controls, and Consequences, Editors: T-F. Lin. IWA Publishing, UK. ISBN13: 9781780406657; eISBN: 9781780406664, 2019, pp 245-280. 

Line 303. Define alkali ion and provide chemical structure. 

Added to the footnote.

Line 314-323. Re-state emphatically that this study is focused on mineral waters of generally >500 mg/L mineral residue or TDS (see Table 2) and that many waters were very high in calcium carbonate. This is important for interpretation of the data. 

Added.

Methods 

Line 328 and forward. Define the hedonic scale: is 1 a like or dislike?  

The evaluators assessed the overall taste on a 10 cm graphic unstructured scale with a border of both sides, in which one extreme was marked "disgusting" and the other as "excellent". The graphical representation on the scale was converted to values with a ruler to the nearest half-centimeter.

Results  

Table 2. Remove “very poorly” as header as no waters taste-tested have < 50 mg/L mineral residue. There are not waters in this group. 

Waters were split into four groups:

  • VPM = Very poorly mineralized – with dissolved substances up to 50 mg/l,
  • PM = Poorly mineralized – content of solutes 50 to 500 mg/l,
  • MM = Moderately mineralized – content of solutes 500 to 1500 mg/l,
  • SM = Strongly mineralized – content of solutes 1500 to 5000 mg/l,
  • VSM = Very strongly mineralized – the content of solutes greater than 5000 mg/l.

Table 2. While the authors did not measure TDS nor sulfate (a major TDS ion in some waters), it would be helpful for the reader to have a sense of TDS. A suggestion is to add a column to table 2 that is the “Sum of measured Minerals” which will provide an indicator of the overall mineral content of the waters. This will not be TSD, but it will provide perspective. If the water producer/bottler provided TDS as a parameter on their product, then provide the value. 

Added a column with sum of measured minerals. Data from water producer are not available.

Discussion 

Summarize data in Table 2 to indicate that most of the high mineral content waters in this study had substantial calcium and carbonate and little chloride. This is what distinguishes this study from other drinking water taste investigations and provides novelty. Most municipal drinking waters strive for TDS< 500 mg/L; these waters are typically >500 mg/L TDS. Chloride<

Added a paragraph with highlighting the differences with other articles and added the comparison.

Lines 458-460. An important aspect of consumer behavior toward drinking water is their perception of the water relative to what they typically drink and their liking of mineral tastes. Consumers like similar and familiar products; Teillet et al. address this. The role of consumer preference in the mineralization of water is similar to that of other consumer products: for example, some consumers like crackers with high salt (NaCl) and a salty taste, others like low salt and less salty taste. 

Agreed and added.

Line 471. Whelton et al. 2007 is not the correct reference as their article is a review and does not have experimental data. 

Updated wording to emphasise, that it is a review, not experimental article.

Conclusions 

The conclusions are very general and can be improved by added specific and number values. 

After discussion among authors, we decided to keep conclusion in the form of concluding summary, which we feel is easier to be digested by readers.

Round 2

Reviewer 1 Report

Much improved.

Author Response

Dear Reviewer 1, 

thank you for all the comments and for recommending our paper for acceptance. 

Regards,

Hynek Roubík

Reviewer 2 Report

The authors improved the manuscript as asked.

I don't have any other comment.

Author Response

Dear Reviewer 2, 

thank you for all the comments and for recommending our paper for acceptance. 

Regards,

Hynek Roubík

Reviewer 3 Report

Honig et al. R1.

The authors incorporated the suggestions of the three reviewers to improve the manuscript.  A few more corrections are required.

Table 3 has a Taste rated with numerical values so the Methods must explain and provide numerical values. A suggestion is to add “0” and “10” to section 3.2.3, such as “marked "disgusting (0)" and the other as "excellent" (10).

Ln 505-506.  Reword sentence. “Scarce” is not correct. Is it is not possible to investigate individual nutrients (e.g., Ca2+, Mg2+,  Na+) because they must be added as a salt with a cation and anion.  “mainly because studies investigating the actual effect of individual nutrients on taste are scarce.”

References required updating, as they do not agree.  There are numbers 1-62 for references in the text and only 58 references listed in the reference section.

Author Response

Dear Reviewer 2,

thank you for all the additional comment.

We have improved our paper accordingly. 

Table 3 has a Taste rated with numerical values so the Methods must explain and provide numerical values. A suggestion is to add “0” and “10” to section 3.2.3, such as “marked "disgusting (0)" and the other as "excellent" (10).

-Changed. 

Ln 505-506.  Reword sentence. “Scarce” is not correct. Is it is not possible to investigate individual nutrients (e.g., Ca2+, Mg2+,  Na+) because they must be added as a salt with a cation and anion.  “mainly because studies investigating the actual effect of individual nutrients on taste are scarce.”

-as the sentence was not necessary, we deleted the sentence for better clarity. 

References required updating, as they do not agree.  There are numbers 1-62 for references in the text and only 58 references listed in the reference section.

-thank you for noticing. The references are now correctly placed in the References section.

Regards,

Hynek Roubík